# Understanding Emergent Abilities of Language Models from the Loss Perspective

**Zhengxiao Du**[1,2], **Aohan Zeng**[1,2], **Yuxiao Dong**[2], **Jie Tang**[2]
[1]Zhipu AI [2]Tsinghua University
{zx-du20,zah22}@mails.tsinghua.edu.cn

## Abstract

Recent studies have put into question the belief that emergent abilities [58] in language models are exclusive to large models. This skepticism arises from two observations: 1) smaller models can also exhibit high performance on emergent abilities and 2) there is doubt on the discontinuous metrics used to measure these abilities. In this paper, we propose to study emergent abilities in the lens of pre-training loss, instead of model size or training compute. We demonstrate that the Transformer models with the same pre-training loss, but different model and data sizes, generate the same performance on various downstream tasks, with a fixed data corpus, tokenization, and model architecture. We also discover that a model exhibits emergent abilities on certain tasks—regardless of the continuity of metrics—when its pre-training loss falls below a specific threshold. Before reaching this threshold, its performance remains at the level of random guessing. This inspires us to redefine emergent abilities as those that manifest in models with lower pre-training losses, highlighting that these abilities cannot be predicted by merely extrapolating the performance trends of models with higher pre-training losses.

## 1 Introduction

Scaling of language modes (LMs) on both model and data sizes has been shown to be effective for improving the performance on a wide range of tasks [42, 6, 23, 8, 65, 55, 36], leading to the widespread adoption of LM applications, e.g., ChatGPT. The success of such scaling is guided by scaling laws [22, 28, 10, 23], which study the predictability of pre-training loss given the model and data sizes.

While scaling laws focus on the pre-training loss, the scaling effect on the performance of downstream tasks has thus far less studied. Emergent abilities [58] are defined as abilities that present in larger LMs but not present in smaller one. The existence of such abilities is recently challenged for two reasons. First, small LMs trained on a sufficient amount of high-quality data can outperform large models on tasks with claimed emergent abilities [55, 56, 26]. For example, LLaMA-13B with less compute [55] can outperform GPT-3 (175B) on MMLU [21], due to more training tokens and improved data-filtering methods. Second, Schaeffer et al. [46] claim that emergent abilities appear due to the nonlinear or discontinuous metrics selected to evaluate certain datasets, rather than from a fundamental change in larger models.

The Chinchilla scaling laws [23] show that different combinations of model sizes and data sizes can lead to different pre-training losses even with the same training compute. Consequently, the pre-training loss can naturally better represent the learning status of LMs than the model or data sizes. However, the relationship between the loss of an LM and its performance on downstream tasks is not yet well understood. Existing literature has either focused on the transfer learning paradigm [33, 54] or constrained its study to single models, tasks, or prompting methods [49, 61].

38th Conference on Neural Information Processing Systems (NeurIPS 2024).

In this work, we propose to study emergent abilities from the perspective of pre-training loss instead of model size or training compute. To examine the relationship between the pre-training loss of LMs and their performance, we pre-train more than 30 LMs of varied model and data sizes from scratch, using a fixed data corpus, tokenization, and model architecture. Their downstream performance is evaluated on 12 diverse datasets covering different tasks, languages, prompting types, and answer forms. We demonstrate that the pre-training loss of an LM is predictive of its performance on downstream tasks, regardless of its model size or data size. The generality of this conclusion is further verified by extracting and observing the performance and loss relationship of the open LLaMA [55] and Pythia [3] models.

Over the course, we find that performance on certain downstream tasks only improves beyond the level of random guessing when the pre-training loss falls below a specific threshold, i.e., emergent abilities. Interestingly, the loss thresholds for these tasks are the same. When the loss is above this threshold, performance remains at the level of random guessing, even though performance on other tasks continues to improve from the outset. To exclude the impact of discontinuous metrics [46, 61], we evaluate the emergent performance increase using continuous metrics and show that the emergent abilities persist across both discontinuous and continuous metrics.

Based on these observations, we define the emergent abilities of LMs from the perspective of pre-training loss: an ability is emergent if it is not present in language models with higher pre-training loss, but is present in language models with lower pre-training loss. According to the loss scaling laws [22, 28], the pre-training loss is a function of model size, data size, and training compute. Therefore, the new emergent abilities can also account for the previously-observed emergent abilities in terms of model size or training compute.

The advantage of the new definition lies in its ability to better capture the tipping points in training trajectories when LMs acquire emergent abilities. Once again [58], the existence of emergent abilities suggests that we cannot predict all the abilities of LMs by simply extrapolating the performance of LMs with higher pre-training loss. Further scaling the model and data size to lower the pre-training loss may enable new abilities that were not present in previous LMs.

## 2   Does Pre-training Loss Predict Task Performance?

Table 1: English and Chinese datasets evaluated in the experiment, and their task types, prompting types, answer forms and metrics. For prompting type, we refer to the chain-of-thought prompting [59] as few-shot CoT and the original in-context learning prompting [6] as few-shot.

| Dataset | Task | Prompting Type | Answer Form | Metric |
|---------|------|----------------|-------------|--------|
| *English datasets* | | | | |
| TriviaQA [27] | Closed-book QA | Few-shot | Open-formed | EM |
| HellaSwag [64] | Commonsense NLI | Zero-shot | Mulit-choice | Accuracy |
| RACE [31] | Reading Comprehension | Few-shot | Multi-choice | Accuracy |
| WinoGrande [44] | Coreference Resolution | Zero-shot | Multi-choice | Accuracy |
| MMLU [21] | Examination | Few-shot | Multi-choice | Accuracy |
| GSM8K [12] | Math Word Problem | Few-shot CoT | Open-formed | EM |
| *Chinese datasets* | | | | |
| NLPCC-KBQA[15] | Closed-book QA | Few-shot | Open-formed | EM |
| ClozeT [63] | Commonsense NLI | Zero-shot | Multi-choice | Accuracy |
| CLUEWSC [62] | Coreference Resolution | Zero-shot | Multi-choice | Accuracy |
| C3 [52] | Reading Comprehension | Few-shot | Multi-choice | Accuracy |
| C-Eval [25] | Examination | Few-shot | Multi-choice | Accuracy |
| GSM8K-Chinese | Math Word Problem | Few-shot CoT | Open-formed | EM |

We study the relationship between the performance of the language models (LMs) on 12 downstream tasks and the pre-training loss. We pre-train LMs of different model sizes (300M, 540M, 1B, 1.5B, 3B, 6B, and 32B) on varied numbers of tokens with fixed data corpus, tokenization, and architecture. In addition, we leverage the open LLaMA [55] models (7B, 13B, 33B, and 65B) to validate our observations.

It is not straightforward that the loss of LMs decides the performance on downstream tasks. Generally the performance is decided by the probability to predict the ground truth $y$ given the prompt $x$, i.e. $p(y|x)$. The probability can be written as a function of the cross entropy loss:

$$p(y|x) = \exp(-\ell(y|x)) \tag{1}$$

where $\ell(y|x)$ is the cross entropy loss of the LM given the context $x$ and the target $y$. While $\ell(y|x)$ has the same form as the pre-training loss $L$, they are not equal. First, the pre-training loss is an average of all the tokens in all the documents pre-trained on. According to our empirical observation, the losses of different documents are not uniform. Second, if $x$ and similar documents do not exist in the pre-training corpus, $\ell(y|x)$ is the generalization loss, which is often related to other factors beyond the training loss, such as the model size. For example, in computer vision, a highly over-parameterized models often improve over an under-parameterized models in test performance when both models converge on the training data [14, 7].

## 2.1 Pre-training Setting

All the models are pre-trained on a mixture of English and Chinese corpus. The ratio of English to Chinese is 4:1 in the pre-training corpus. The model architecture is similar to LLaMA [55] with two differences: we use grouped-query attention [1] to replace the multi-query attention and we apply rotary position embedding on half the dimensions of the query and key vectors. More details can be found in Appendix A.

## 2.2 Evaluation Tasks

To present a comprehensive demonstration, we evaluate the pre-trained models on 12 datasets across different tasks and prompting types in both English and Chinese. The six task types include:

**Closed-book QA:** Answering questions about the real world based solely on the pretrained knowledge. We use TriviaQA [31] for English. For Chinese, we build a closed-book QA dataset based on NLPCC-KBQA [15] dataset following the TriviaQA format.

**Commonsense Natural Language Inference (NLI):** Selecting the most likely followup given an event description. We use the HellaSwag dataset [64] for English and the ClozeT dataset in Yao et al. [63] for Chinese.

**Reading comprehension:** Reading a given article or paragraph and answering questions about it. We use RACE [31] for English and C3 [52] for Chinese. Both are based on multi-choice questions.

**Coreference Resolution:** Given a sentence with pronouns, determine which pronoun refers to which entity. We use the WinoGrande dataset [44] for English and the CLUEWSC dataset [62] for Chinese.

**Examination:** Multiple-choice questions in examinations. For English, we use MMLU [21], which includes mathematics, US history, computer science, law, and more. For Chinese, we use C-Eval [25] which ranges from humanities to science and engineering.

**Math Word Problem**: Solving real-life, situational and relevant problems using mathematical concepts. For English we use the GSM8K [12] dataset. For Chinese, we translate the questions and answers in GSM8K to Chinese, namely GSM8K-Chinese.

The prompting types cover few-shot [6], zero-shot, and few-shot chain-of-thought (CoT) [59]. The datasets are summarized in Table 1.

## 2.3 Pre-training Loss vs. Performance

In the first experiment, we train three models with 1.5B, 6B, and 32B parameters and observe their behaviors until trained on 3T, 3T, and 2.5T tokens, respectively. The training hyperparameters are shown in Table 4 (Appendix).

We evaluate the performance of intermediate training checkpoints. The checkpoints are saved around every 43B tokens during pre-training. We plot the points of task performance ($y$-axis) and training

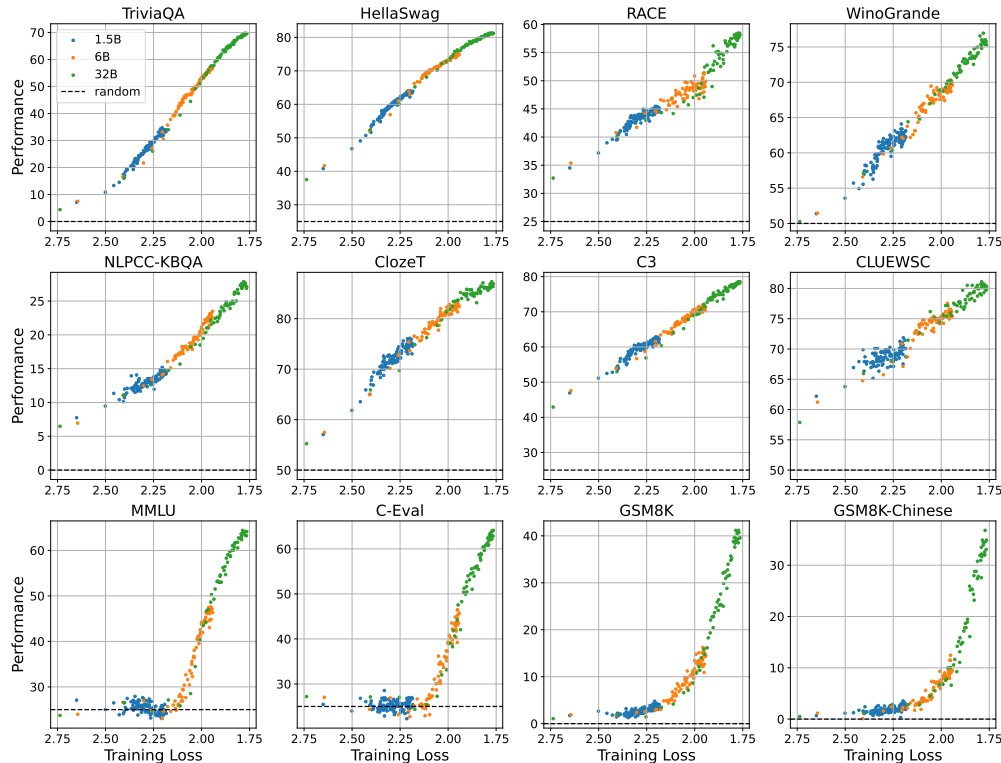

Figure 1: **The performance-vs-loss curves of 1.5B, 6B, and 32B models.** Each data point is the loss ($x$-axis) and performance ($y$-axis) of the intermediate checkpoint of one of the three models. We mark the results of random guess in black dashed lines.

Table 2: Statistical measures of the correlation between task performance and pre-training loss in Figure 1. The spearman correlation coefficient [50] measures the monotonicity of the relationship between the two variables, and the pearson correlation coefficient measures the linearity of the relationship. Both vary between -1 and +1 with 0 implying no correlation. Correlations of -1 or +1 imply an exact monotonic/linear relationship.

| Dataset | TriQA | HS | RACE | WG | NQA | ClozeT | C3 | CW | MMLU | CE | GSM | GSMC |
|---|---|---|---|---|---|---|---|---|---|---|---|---|
| Spearman | -0.996 | -0.996 | -0.977 | -0.978 | -0.984 | -0.986 | -0.988 | -0.947 | -0.804 | -0.831 | -0.975 | -0.948 |
| Pearson | -0.994 | -0.994 | -0.963 | -0.988 | -0.982 | -0.985 | -0.993 | -0.972 | -0.903 | -0.884 | -0.874 | -0.829 |

loss ($x$-axis) in Figure 1, and provide the statistical measures of the two variables in Table 2. From the curves and statistics, we can see that the training loss is a good predictor of the performance on 12 downstream tasks.

- Generally, the task performance improves as the training loss goes down, regardless of the model sizes. On MMLU, C-Eval, GSM8K, and GSM8K-Chinese, all models of three sizes perform at the random level until the pre-training loss decreases to about 2.2, after which the performance gradually climbs as the loss decreases. Detailed analysis on this is shown in Section 3.

- Importantly, the performance-vs-loss data points of different model sizes fall on the same trending curve. That is, by ignoring the color differences (model sizes), the data points of different models are indistinguishable. For example, when the training loss falls around 2.00, the green and orange points on TriviaQA are indistinguishable. This indicates that the model performance on downstream tasks largely correlates with the pre-training loss, *regardless of the model size.*

- Both spearman and pearson correlation coefficients show that performance is strongly related to pre-training loss on TriviaQA, HellaSwag, RACE, WinoGrande, etc. The pearson correlation coefficients on these tasks specifically show that points from different models lie on the same trending curve. The relationship is weaker on MMLU, CEval, GSM8K, and GSM8K-Chinese, verifying the emergence of performance which we discuss in Section 3.

- Interestingly, we find that the overall training loss is a good predictor of performance on both English and Chinese tasks, although it is computed on a mixture of English and Chinese tokens. This implies that the learning dynamics of English and Chinese tokens are likely very similar during multilingual pre-training.

## 2.4 Training Token Count vs. Performance

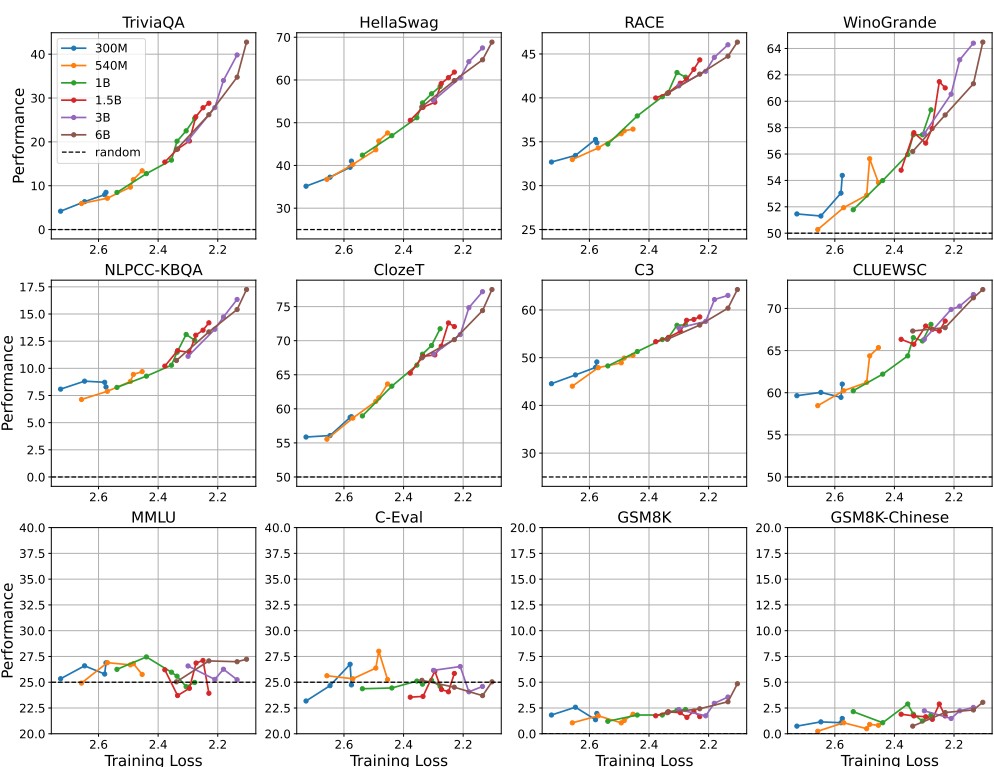

Figure 2: **The performance-vs-loss curves of smaller models pre-trained with different numbers of training tokens**. Each data point is the loss ($x$-axis) and performance ($y$-axis) of the final checkpoint of one model, i.e., each point corresponds to one model trained from scratch. We mark the results of random guess in black dashed lines.

Following the empirical experiments in scaling laws [22, 28, 23], we further pre-train 28 relatively smaller models with different numbers of training tokens. The model sizes range from 300M, to 540M, 1B, 1.5B, 3B, and to 6B, while the numbers of pre-training tokens range from 33B to 500B. Varying the number of pre-training tokens is necessary since to achieve optimal performance we need to set the cosine learning rate schedule to reach the minimum at the corresponding token count [28, 23]. The number of tokens used and hyperparameters for all models are shown in Table 5 (Appendix).

On each line, each data point represents the performance and pre-training loss of the corresponding model pre-trained completely from scratch with the certain token count (and learning rate schedule). We can see that similar to the observations from Figure 1, the data points of different models sizes and training tokens largely fall on the same trending curves. In other words, *the LMs with the same pre-training loss regardless of token count and model size exhibit the same performance on the 12 downstream tasks.*

Another similar observation is that the performance curves on MMLU, C-Eval, GSM8K, and GSM8K-Chinese do not yield an uptrend, meaning that the performance of these models on these four tasks are close to random (with fewer than 500B tokens). For simplicity, we only plot the performance of the latest checkpoint in each training in Figure 2. The complete performance curves with intermediate checkpoints of each model, in which we can observe the same trend but larger variance, are shown in Figure 5 (Appendix).

## 2.5 LLaMA's Loss vs. Performance

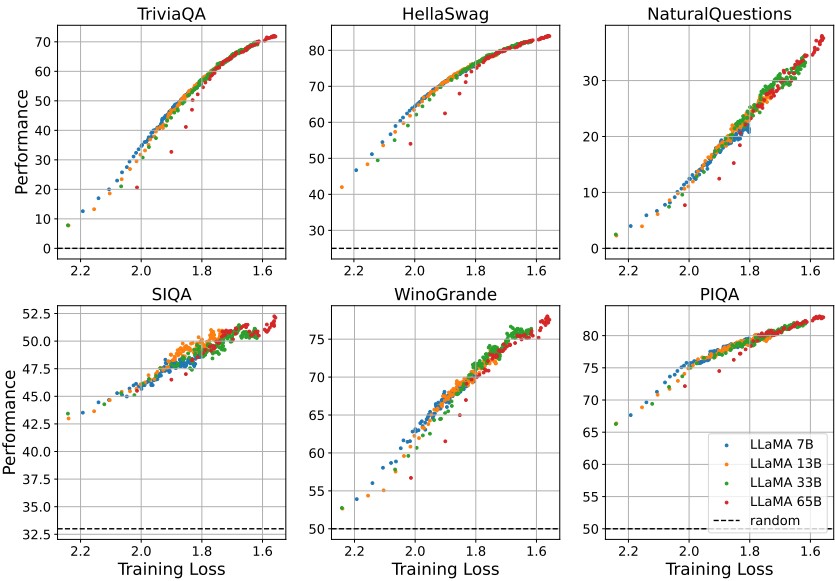

Figure 3: **The performance-vs-loss curves of LLaMA.** The values of performance and training loss are extracted from the figures in the original LLaMA paper [55]. Note that the LLaMA2 paper [56] does not cover such figures with related information.

To validate the generality of our observations, we analyze two different model series with required information made publicly available, i.e., LLaMA [55] and Pythia [3]. Compared to our models, LLaMA uses a pre-training corpus that excludes Chinese documents, leverages a different pre-training framework [37], and adopts a slightly different model architecture. Since the intermediate checkpoints of LLaMA are not available, we extract the pre-training loss and corresponding performance on six question answering and commonsense reasoning tasks from the figures in its original paper, and plot the points in Figure 3.

Excitingly, most data points from the LLaMA models with different sizes (7B, 13B, 33B, 65B) fall on the same upwards trend. This observation further confirm our conclusion that the model's pre-training loss can predict its performance on downstream tasks, regardless of model size and token count. Note that there is only one exception at the early stage of LLaMA-65B. We can see that when the training loss is higher than 1.8, LLaMA-65B performs worse than smaller models with the same training loss. Without access to its intermediate checkpoints, we unfortunately cannot further analyze the result. One possible explanation is that they use exponential smoothing on either the loss or downstream performance plots. Exponential smoothing would perturb the earlier points more than other points, potentially leading to this effect. Note that the outliers only constitute the initial 10% training tokens. The results for Pythia are shown in Appendix F, which also support our conclusion.

Observed from previous experiments and analysis, we can conclude that the pre-training loss is a good indicator of LMs' performance on downstream tasks, independent of model sizes, training tokens, languages, and pre-training frameworks.

## 3 Analysis of Different Tasks and Metrics

### 3.1 Performance Trends of Different Tasks

In Figures 1 and 2, we can separate the datasets into two groups: First, on TriviaQA, HellaSwag, RACE, WinoGrande, NLPCC-KBQA, ClozeT, CLUEWSC, and C3, the performance improves smoothly with decreased pre-training loss from the very beginning. Second, on MMLU, C-Eval, GSM8K, and GSM8K-Chinese, the performance remains flat when the loss is higher than a certain threshold. Once the pre-training loss is lower than this threshold, the performance starts to improve.

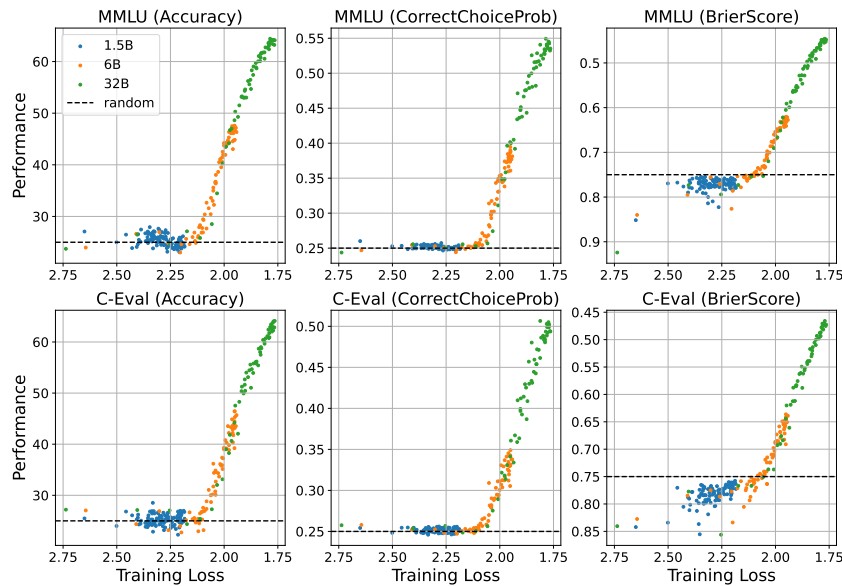

Figure 4: **The performance-vs-loss curves of different metrics on MMLU and C-Eval.** Accuracy: discontinuous; CorrectChoiceProb and BrierScore: continuous. We mark the result of random guess in black dashed lines.

The correlation coefficients in Table 2 also reveal the difference: coefficients in the first group are close to -1, indicating strong correlations, while correlations in the second group are weaker.

Take MMLU as an example of the second group, when the pre-training loss is higher than 2.2, the accuracy remains around 25%. Since each question in MMLU has four options, this means the model prediction is no better than random guessing. However, when the pre-training loss drops below 2.2, the accuracy increases as the loss decreases, similar to the trend observed in the first group of tasks. The performance trends of C-Eval, GSM8K, and GSM8K-Chinese follow a similar pattern. Despite differences in languages, tasks, prompting types, and answer forms among the four datasets are different, the thresholds for performance improvement are surprisingly all around 2.2.

RACE in the first group has a prompting format similar to MMLU: both consist of multi-choice examination questions with in-context demonstrations, but their performance curves are quite different. We hypothesis that it is the task difficulty that makes the difference. Tasks of the first group of datasets are easier than those of the second group. For example, RACE requires the model to select correct answers for questions about a given article, and HellaSwag lets the model to select the possible followup of a situation based on commonsense. In contrast, MMLU and C-Eval consist of questions designed for high school, college, or professional examinations, requiring a broader range of knowledge. GSM8K and GSM8K-Chinese are math word problems that were previously considered as impossible to be solved by pre-trained language models before Chain-of-Thought prompting.

The phenomenon can be related to grokking, which describes the improvement of performance from the random chance level to perfect generalization [40]. Power et al. [40] find that this improvement can occur well past the point of overfitting. In pre-training, the models are usually underfitting instead of overfitting overall. Since the pre-training corpus is a mixture of different documents, it is possible that the model already fits some patterns—such as numerical addition—in the data, while still underfitting the overall corpus.

Certainly, the observations on the second groups of datasets can also be related to emergent abilities [58], that is, abilities that only present in large models. According to the scaling law [28], with the number of training tokens fixed, the pre-training loss follows a power law with respect to model sizes. In other words, there is a monotonic relationship between model size and pre-training loss. For the second group of tasks, there is a threshold of model sizes that corresponds to the tipping point in the pre-training loss. When the model size exceeds this threshold, the model can exhibit performance above the random chance level.

## 3.2 Influence of Different Metrics

Schaeffer et al. [46] propose an alternative explanation of emergent abilities of LMs, that is, emergent abilities appear due to the researchers' choice of nonlinear or discontinuous metrics. The accuracy on multi-choice questions (e.g., MMLU) is discontinuous, since the score on a question is either 1 or 0. To validate this claim, we examine the intermediate checkpoints on MMLU and C-Eval with continuous metrics rather than discontinuous accuracy used in the original benchmarks. The first metric is the predicted probability of the correct answer, denoted as CorrectChoiceProb. The second one is the Brier Score [5] used in Schaeffer et al. [46]:

$$\text{BrierScore} = \frac{1}{N} \sum_{i=1}^{N} \sum_{j=1}^{C} (y_{ij} - \hat{y}_{ij})^2 \tag{2}$$

where $\hat{y}_{ij}$ is the predicted probability of sample $i$ for class $j$ and $y_{ij}$ is the ground truth probability. The metric measures the prediction error and a lower value indicates better performance.

We plot the results measured by different metrics on MMLU and C-Eval in Figure 4. All three metrics—accuracy, correct choice probability, and Brier Score—show emergent performance improvements (value increase for the first two and decrease for the third) when the pre-training loss drops below a certain threshold. The Brier Score also decreases when the pre-training loss is above the threshold. However, the decrease of Brier Score does not always represent improvements on the task, since the Brier Score is related to not only the predicted probability of the correct answer but also the predicted probabilities of the incorrect answers. We find that the distribution of the correct answers is uniform in the four options in MMLU and C-Eval. The best Brier Score for a context-free predictor is achieved by always giving uniform probability to all the options. In this case, the Brier Score is equal to 0.75. Therefore, the performance in terms of Brier Score is no better than random guess before the loss reaches the threshold. This observation further confirms our previous conclusion. We discuss the contrary observations of Schaeffer et al. [46] and Xia et al. [61] in Appendix C.

We conclude that emergent abilities of language models occur when the pre-training loss reaches a certain tipping point, and continuous metrics cannot eliminate the observed tipping point.

## 4 Defining Emergent Abilities from the Loss Perspective

In previous sections, we show that 1) the pre-training loss is predictive of the performance of language modes on downstream tasks, and 2) some tasks exhibit emergent performance improvements from the random guess level when the pre-training loss drops below a certain threshold regardless of model size, token count, and continuity of metrics. Based on these observations, we give a new definition of emergent abilities from the pre-training loss perspective:

**Definition.** *An ability is emergent if it is not present in models with higher pre-training loss but is present in models with lower pre-training loss.*

The normalized performance on an emergent ability as a function of the pre-training loss $L$ is:

$$\begin{cases} f(L) & \text{if } L < \eta \\ 0 & \text{otherwise} \end{cases} \tag{3}$$

where $f(L)$ is a monotonically decreasing function of $L$, $\eta$ is the threshold, and the normalized performance of random guess is 0.

Next we will show how the new definition can be related to previously observed emergent abilities [58]. In Henighan et al. [22], they give the scaling relation for the loss with model size $N$ when the number of training tokens $D$ is fixed:

$$L(N) = L_\infty + \left(\frac{N_0}{N}\right)^{\alpha_N} \tag{4}$$

where $L_\infty$ is the irreducible loss, and $\alpha_N$ is the coefficient. The equation shows that the loss of language models follows a power-law plus a constant. Combining Equation (3) and Equation (4), we can get the normalized performance as a function of the model size $N$

$$\begin{cases} f\left(L_\infty + \left(\frac{N_0}{N}\right)^{\alpha_N}\right) & \text{if } N \geq N_0 \cdot (\eta - L_\infty)^{-\frac{1}{\alpha_N}} \\ 0 & \text{otherwise} \end{cases} \tag{5}$$

From this equation, we can explain the emergent abilities observed in Wei et al. [58]: when model sizes are smaller than $N_0 \cdot (\eta - L_\infty)^{-1/\alpha_N}$, the normalized performance is zero. When model sizes exceed $N_0 \cdot (\eta - L_\infty)^{-1/\alpha_N}$, the increase in model size leads to a decrease of pre-training loss and an improvement in normalized performance.

## 5    Related Work

**Relationship of Pre-training Loss and Task Performance.** In the transfer learning setting, i.e. the language model is pre-trained on the general corpus and fine-tuned on supervised data of specific tasks, Tay et al. [54] find that models with the same pre-training loss can have different downstream performance after finetuning, due to inductive bias in model architectures such as Transformers and Switch Transformers. Tay et al. [53] further study the effect of model shapes on downstream fine-tuning. Liu et al. [33] also study the effect of inductive bias of model sizes and model algorithms on the relationship of pre-training loss and downstream performance after fine-tuning, but their theory only applies in the saturation regime, where the models are close to minimal possible pre-training loss. Instead, large language models today are generally under-trained [23, 55], far from the saturation regime. Overall, these studies focus on the pretrain-finetune paradigm, in which inductive bias helps improve transferability, while we study prompted performance of large language models without finetuning [29, 6]. For the prompted performance of large language models, Xia et al. [61] claim that perplexity is a strong predictor of in-context learning performance, but the evidence is limited to the OPT model [66] and a subset of BIG-Bench [51]. Instead, Shin et al. [49] find that low perplexity does not always imply high in-context learning performance when the pre-training corpus changes. Gadre et al. [18] fits the relation of perplexity and the top-1 error averaged over many natural language tasks with a power law. Instead, we focus on the different relations of tasks and a small part of tasks that show emergency trends.

**Emergent abilities.** Wei et al. [58] propose the idea of emergent abilities, abilities that only present in large language models. This is similar to the claim of Ganguli et al. [19] that it is more difficult to predict the capacities of language models than to predict the pre-training loss. The existence of emergent abilities has been challenged. Hoffmann et al. [23] show that smaller language models trained with sufficient data can outperform undertrained larger language models, supported by follow-up models [55, 26, 56]. On the other hand, Schaeffer et al. [46] claim that emergent abilities are due to the discontinuous metrics used for evaluation, also found in Xia et al. [61]. Similarly, Hu et al. [24] propose to predict the performance of emergent abilities with the infinite resolution evaluation metric. In this paper we prove the existence of emergent abilities from the perspective of pre-training loss, even with continuous metrics.

## 6    Conclusion

Our paper proposes a new definition of emergent abilities of language models from the perspective of pre-training loss. Empirical results show that the pre-training loss is a better metric to represent the scaling effect of language models than model size or training compute. The performance of emergent abilities exhibits emergent increase when the pre-training loss falls below a certain threshold, even when evaluated with continuous metrics.

The new definition offers a precise characterization of the critical junctures within training trajectories where emergent abilities manifest. It encourages future studies to investigate the shifts in language models at these junctures, which facilitate the development of new capabilities.

## 7    Limitation

We study the relationship of pre-training loss and task performance across model sizes, training tokens, tasks, languages, prompting types, and answer forms. Factors we have not considered are model architectures and training algorithms. We analyze the performance-loss curves of LLaMA and Pythia with slightly different architectures, and find that the relationship holds for all the models. But there are fundamentally different model architectures, such as routed Transformers [16], and

non-Transformer architectures [17, 39] beyond our consideration. Both our models and LLaMA use AdamW optimizer [35], while there are other optimizers for language model pre-training [48, 32].

The disadvantage of studying emergent abilities in the lens of pre-training loss is that the pre-training loss is affected by the tokenizer and the distribution of pre-training corpus. The values of pre-training loss of language models trained on different corpus are not directly comparable. One possible solution is to evaluate different language models on a public validation set with the normalized perplexity [43] to account for the different vocabulary sizes.

The paper should not be considered as a push to expand model sizes and data sizes of language models beyond current scales. It is not guaranteed that new tipping points emerge in larger scales. Also, instruction tuning [57, 45, 9, 34] can improve the zero-shot performance of language models on unseen tasks, including MMLU and GSM8K.

## Acknowledgments and Disclosure of Funding

This work is supported by the Natural Science Foundation of China NSFC 62425601 and 62276148, a research fund from Zhipu, New Cornerstone Science Foundation through the XPLORER PRIZE and Tsinghua University (Department of Computer Science and Technology)-Siemens Ltd., China Joint Research Center for Industrial Intelligence and Internet of Things (JCIIOT). Corresponding authors: Yuxiao Dong and Jie Tang.

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

# A  Pre-training Settings

## A.1  Pre-training Corpus

| Source | Ratio |
|---|---|
| CommonCrawl | 80.2% |
| Code | 10.0% |
| Books | 3.8% |
| Wikipedia | 3.8% |
| Papers | 1.6% |
| StackExchange | 0.6% |

Table 3: The ratio of different sources in the English corpus.

Our pre-training corpus is a mixture of English and Chinese documents. The ratio of English tokens to Chinese tokens in the pre-training corpus is 4:1. Both the English and Chinese corpora consist of webpages, wikipedia, books, and papers. The distribution of different sources in the English corpus is shown in Table 3. The distribution and processing pipeline are similar to Redpajama [13]. During pre-training, documents from Wikipedia and Books are trained for multiple epochs, but most documents (93.4% in the pre-training corpus) are never repeated.

We tokenize the data with the byte pair encoding (BPE) algorithm [47] in the SentencePiece package [30]. The vocabulary size is 65k.

## A.2  Hyperparameters

The hyperparameters for training of 1.5B, 6B, and 32B models are shown in Table 4. The hyperparameters for training of smaller models are shown in Table 5. The sequence length is 2048 and the optimizer is AdamW [35] with $\beta_1 = 0.9$ and $\beta_2 = 0.95$.

# B  Evaluation Settings

The evaluated splits and numbers of examples are summarized in Table 6. For English datasets, we follow Gopher [41] and Chinchilla [23]'s selection of evaluation splits. For Chinese datasets, we use the validation split when the ground labels are always available. For CLUEWSC, the size of the validation set is too small (100), so we combine the train and validation splits. GSM8K-Chinese is translated from GSM8K with machine translation and human proofreading.

# C  Are Emergent Abilities of Language Models a Mirage?

[46] claim that emergent abilities proposed in [58] are mainly a mirage caused by nonlinear and discontinuos metrics. [61] also support the idea.

[61] use the perplexity of correct options as the metric for BIG-Bench and find that the metric impproves smoothly on almost all the tasks of BIG-Bench. We argue that the perplexity of correct options is not the correct metric to evaluate the performance of multi-choice questions. The correct metric of multi-choice questions should reflect the ability of distinguishing correct options from incorrect options. The perplexity of correct options and incorrect options may decrease simultaneously. In fact, [61] already observe perplexity of incorrect options decreasing during pre-training and only at

| Parameters | Tokens | d_model | d_hidden | n_heads | n_layers | Batch Size | Max LR |
|---|---|---|---|---|---|---|---|
| 1.5B | 3T | 2048 | 6912 | 16 | 24 | 1344 | 5e-4 |
| 6B | 3T | 4096 | 13696 | 32 | 28 | 4224 | 4e-4 |
| 32B | 2.5T | 6656 | 22272 | 52 | 58 | 8832 | 3e-4 |

Table 4: Hyperparameters of pre-training of 1.5B, 6B, and 32B models.

| Parameters | Tokens | d_model | d_hidden | n_heads | n_layers | Batch Size | Max LR |
|---|---|---|---|---|---|---|---|
| 300M | 67B | 1152 | 3840 | 9 | 12 | 1152 | 2.8e-3 |
| 300M | 125B | 1152 | 3840 | 9 | 12 | 1152 | 2.8e-3 |
| 300M | 250B | 1152 | 3840 | 9 | 12 | 1152 | 2.8e-3 |
| 300M | 500B | 1152 | 3840 | 9 | 12 | 1152 | 2.8e-3 |
| 540M | 33B | 1536 | 5120 | 12 | 12 | 1152 | 2e-3 |
| 540M | 66B | 1536 | 5120 | 12 | 12 | 1152 | 2e-3 |
| 540M | 125B | 1536 | 5120 | 12 | 12 | 1152 | 2e-3 |
| 540M | 250B | 1536 | 5120 | 12 | 12 | 1152 | 2e-3 |
| 540M | 500B | 1536 | 5120 | 12 | 12 | 1152 | 2e-3 |
| 1B | 33B | 2048 | 6912 | 16 | 16 | 1152 | 1.5e-3 |
| 1B | 67B | 2048 | 6912 | 16 | 16 | 1152 | 1.5e-3 |
| 1B | 125B | 2048 | 6912 | 16 | 16 | 1152 | 1.5e-3 |
| 1B | 250B | 2048 | 6912 | 16 | 16 | 1152 | 1.5e-3 |
| 1B | 500B | 2048 | 6912 | 16 | 16 | 1152 | 1.5e-3 |
| 1.5B | 67B | 2048 | 6912 | 16 | 24 | 1152 | 1e-3 |
| 1.5B | 100B | 2048 | 6912 | 16 | 24 | 1152 | 1e-3 |
| 1.5B | 125B | 2048 | 6912 | 16 | 24 | 1152 | 1e-3 |
| 1.5B | 250B | 2048 | 6912 | 16 | 24 | 1152 | 1e-3 |
| 1.5B | 375B | 2048 | 6912 | 16 | 24 | 1152 | 1e-3 |
| 1.5B | 500B | 2048 | 6912 | 16 | 24 | 1152 | 1e-3 |
| 3B | 67B | 3072 | 10240 | 24 | 24 | 1152 | 7e-4 |
| 3B | 125B | 3072 | 10240 | 24 | 24 | 1152 | 7e-4 |
| 3B | 250B | 3072 | 10240 | 24 | 24 | 1152 | 7e-4 |
| 3B | 500B | 3072 | 10240 | 24 | 24 | 1152 | 7e-4 |
| 6B | 33B | 4096 | 13696 | 32 | 28 | 1152 | 4e-4 |
| 6B | 67B | 4096 | 13696 | 32 | 28 | 1152 | 4e-4 |
| 6B | 125B | 4096 | 13696 | 32 | 28 | 1152 | 4e-4 |
| 6B | 250B | 4096 | 13696 | 32 | 28 | 1152 | 4e-4 |

Table 5: Hyperparameters of pre-training of smaller models. Each line represents one model pre-trained completely from scratch with the certain number of tokens and its corresponding learning rate schedule.

| Dataset | Evaluated Split | Num. Examples |
|---|---|---|
| TriviaQA | validation | 11,313 |
| HellaSwag | validation | 10,042 |
| RACE | test | 4,934 |
| WinoGrande | validation | 1,267 |
| MMLU | test | 14,042 |
| GSM8K | test | 1,319 |
| NLPCC-KBQA | validation | 10,613 |
| ClozeT | validation | 938 |
| CLUEWSC | train & validation | 508 |
| C3 | validation | 3,816 |
| C-Eval | validation | 1,346 |
| GSM8K-Chinese | test | 1,212 |

Table 6: Statistics of evaluation datasets.

the end of training that the perplexity of correct and incorrect options starts to diverge. This supports the existence of emergent abilities.

[46] use Brier Score [5] as the metric for BIG-Bench. We argue that increase in Brier Score does not always represent improvement of performance on the multi-choice task, since Brier Score is also related to the allocation of probabilities for incorrect options. For example, questions in the MMLU dataset have four options (A, B, C, and D) and the frequency of the four options as correct is equal.

Consider two models that give the same probability independent of questions. One model predicts $(1, 0, 0, 0)$ for the four options and the other model predicts $(0.25, 0.25, 0.25, 0.25)$. The Brier Score for the former is 1.5 while the Brier Score for the latter is 0.75. However, both models do not learn the relationship between questions and correct options at all. One can argue that the latter model better fits the distribution of correct options in the dataset, but the improvement is not as large as the different of 1.5 and 0.75. We should consider the Brier Score of 0.75 as the performance of the random guess baseline, and any decrease in Brier Score above 0.75 should not be considered as the real improvement on the task.

In Figure 6 of [46], they evaluate 4 tasks in BIG-Bench with the Brier Score metric and find that the emergent abilities disappear. We hypothesis that they normalize the Brier Score with the number of options in each question, otherwise the Brier Score of 0.25 on the swahili_english_proverbs task is too low for the smallest model. Four tasks have 2, 2, 4, 5 options in each question. The values of Brier Score for random guess baselines on the four tasks are 0.25, 0.25, 0.1875, and 0.16. Only the largest model surpasses the random guess baseline. This also supports the existence of emergent abilities.

## D    Complete Performance-vs-Loss Curves of Smaller Models

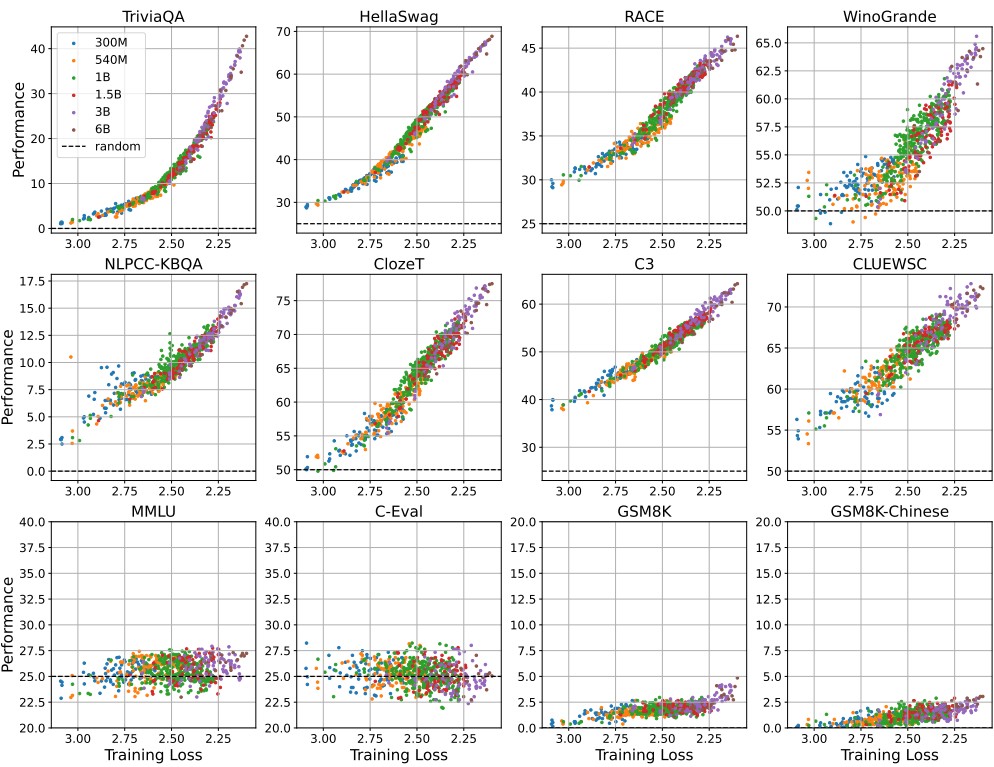

Figure 5: The complete performance-vs-loss curves of smaller models.

The performance-vs-loss curves for all the intermediate checkpoints are shown in Figure 5. The trend is the same as Figure 2, but with larger variance.

## E    Loss vs Compute as an Indicator of Performance

We show the performance-compute curves in Figure 6. Compared with Figure 1, we observe that points from different models do not fall on the same curves on most tasks. This proves that pre-training loss is a better indicator of task performance than compute.

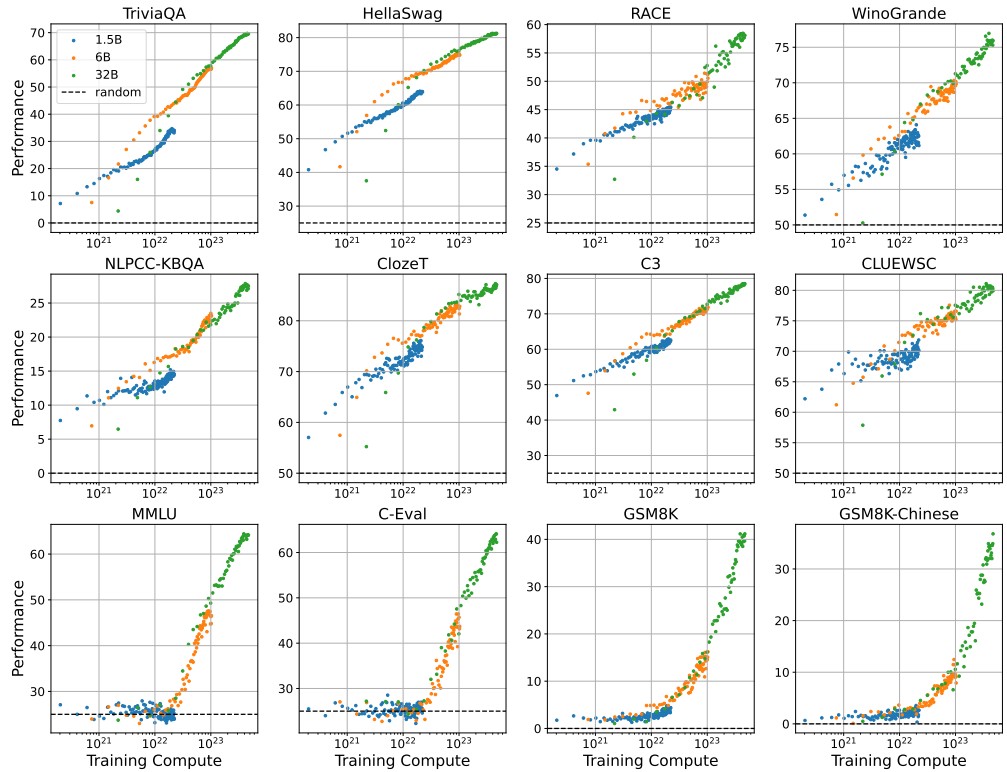

Figure 6: The performance-vs-compute curves of 1.5B, 6B, and 32B models.

# F Pythia's Loss vs. Performance

To further support our conclusion, we plot the performance-loss curves of Pythia [3] in Figure 7. Pythia is a suite of open language models with intermediate checkpoints released. For downstream tasks, we select SciQ [60], LAMBADA [38], WinoGrande [44], ARC-Easy [11], ARC-Challenge [11], and PIQA [4] with reported performance in the official repository. We compute the cross-entropy loss of intermediate checkpoints on the corpus Pile [20]. From the plot, we can observe that the points from different models fall on the same curve on all the tasks. This supports our conclusion that pre-training loss is predictive of task performance.

However, neither Pythia nor LLaMA can be used to analyze the emergent abilities. The largest Pythia model fails to achieve performance above random chance on MMLU and GSM8K [2]. Instead, LLaMA has no intermediate checkpoints released and performance curves on MMLU and GSM8K are not available.

# G Loss vs. Performance on BIG-bench

BIG-bench [51] is a series of diverse tasks designed to evaluate the capacities and limitations of pre-trained language models. Wei et al. [58] find that large language models exhibit emergent abilities on four tasks from BIG-Bench. Among the four tasks, the test set size of the figure-of-speech detection task is too small and the variance is too high. We evaluate the other three tasks in the same setting as Section 2.3 and the results are shown in Figure 8. With pretraining loss decreases along the x-axis, we can clearly observe the tipping point in the performance curves.

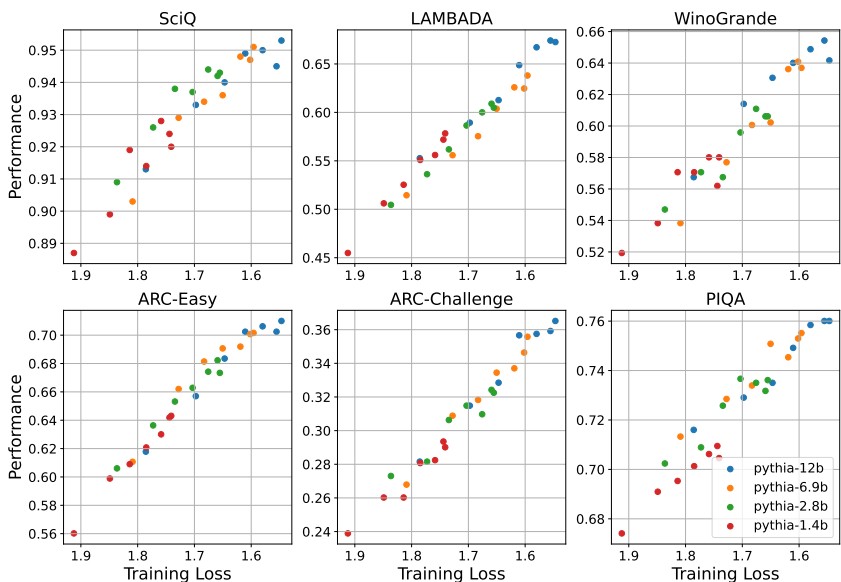

Figure 7: **The performance-vs-loss curves of Pythia.** The performance of Pythia is from the official repository and the loss is evaluated with the released checkpoints.

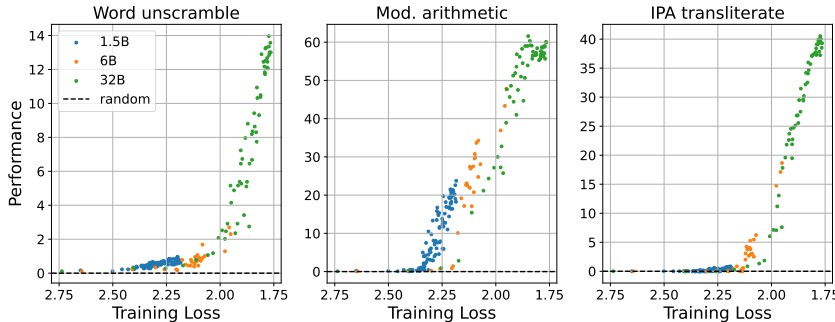

Figure 8: **The performance-vs-loss curves of 1.5B, 6B, and 32B models on 3 tasks in BIG-bench.** Each data point is the loss ($x$-axis) and performance ($y$-axis) of the intermediate checkpoint of one of the three models. We mark the results of random guess in black dashed lines.

## H   Compute Resources

All the models are trained on DGX-A100 GPU (8x80G) servers. The 1.5B, 6B, and 32B models in Section 2.3 take 8 days on 256 A100 GPUs, 8 days on 1024 A100 GPUs, and 20 days on 2048 A100 GPUs respectively. The small models in Section 2.4 take about 20 days on 256 A100 GPUs.

## I   Broader Impact

This paper finds that pre-training loss is predictive of downstream task performance and on some tasks the performance only begins to improve when the pre-training loss falls below a certain threshold. Combined with previous works on scaling laws [28, 22, 23], we can predict the amount of compute required to achieve a certain performance. This can be used to estimate the cost of training a large model.

The paper might encourage companies to expand model sizes and data sizes of language models beyond current scales to pursue new emergent abilities, leading to a waste of compute resources. We want to emphasize that the analysis of previous performance trends do not necessarily apply to the larger models.

