# OpenReview forum: "Understanding Emergent Abilities of Language Models from the Loss Perspective"
_NeurIPS.cc/2024/Conference — NeurIPS 2024 poster_

### Official Review · Reviewer_kB8g · 2024-06-17

**Soundness:** 4
**Presentation:** 4
**Contribution:** 3
**Rating:** 7
**Confidence:** 4

**Summary:**

This paper measures a range of language models' pretraining loss and their downstream performance, arguing that emerging capabilities are better measured by losses as opposed to previously proposed model size or compute (FLOPs). The paper also offers evidence against the argument that emergent abilities are a "mirage" when the task metrics are continuous.

**Strengths:**

1. Clean and comprehensive study on a range of tasks, different models and different model sizes, with a significant number of intermediate checkpoints.
1. Insightful results that builds on top of prior literature: (I) Loss > size and FLOPs and (2) It's not just that downstream task metrics needed to be continuous. Line 226 also makes a good observation on a quirk of Brier Score.

Line 122 makes a thought-provoking claim that many researchers should seriously consider, especially when considering the inference cost of these models during architectural design.
> That is, by ignoring the color differences (model sizes), the data points of different models are indistinguishable… . This indicates that the model performance on downstream tasks largely correlates with the pre-training loss, regardless of the model size.

**Weaknesses:**

1. The authors define emergence as a discontinuity, although it seems possible to fit an exponential function (especially when the x-axis is FLOPs as in Figure 6), or a linear function for many tasks such as the first group in section 3.1. Although Figure 4 offers stronger evidence for discontinuity, it is not definitive, as only MMLU, GSM8K and C-Eval are measured (but I accept that these are reasonably sufficient for an academic paper).
1. The authors discuss why the harder nature of MMLU and GSM8k may be causing the discontinuity, but it would be useful to present a qualitative analysis on the "easier" questions that models "first" get right with just a bit more reduction in pretraining loss.

**Questions:**

1. Is EM the same as perplexity/scoring eval?
1. Why use the training loss as opposed to validation/test loss?

**Limitations:**

L130
> we find that the overall training loss is a good predictor of performance on both English and Chinese tasks,
Naturally, you could compute the per-corpus loss on the x-axis to strength the claim. For the final version, it'd be great to have a per-corpus loss analysis on what kinds of corpora are most predictive of downstream performance.

Not limitations just typos:
Line 128: verifying the emergency of performance -> emergence

Line 218: ground probability -> did you mean ground truth probability?

---

> ### Author Rebuttal · Authors · 2024-08-06
>
> For weakness 1, we add results on three tasks on BIG-Bench in the PDF file of global rebuttal.
>
> For question 1, when the language model directly predicts the answer, EM and perplexity give the same trend. However, with chain-of-thought reasoning, the language model first predicts the intermediate reasoning process then gives the answer. Perplexity evaluation does not work.
>
> For question 2, most of our pretraining corpus go through fewer than one epoch during pretraining and in the early experiments we find that the training loss is consistent with the validation loss. Therefore we use the training loss for simplicity.

---

### Official Review · Reviewer_BK9j · 2024-07-05

**Soundness:** 1
**Presentation:** 2
**Contribution:** 3
**Rating:** 3
**Confidence:** 4

**Summary:**

This paper formulates the concept of emergence as a relationship between language modeling training loss and task performance, rather than in relation to model/data scale.

**Strengths:**

The paper is clear. I appreciate the central message, as descriptions that relate the capabilities strictly to the scale of a language model overlook factors such as data quality and architecture.  If you actually demonstrated that loss was a better predictor than model scale on specific capabilities, the paper would have a lot of value.

The paper is partly making the point of "agreement on the line" https://arxiv.org/abs/2206.13089 that in-distribution loss tends to be strongly correlated with out of distribution or specialized metrics. This point is worth making. The claim that we can identify emergence with respect to loss is more dubious, as the threshold-based measurements used are overly generous to claims of emergence in general.

**Weaknesses:**

Even when different models have very similar pretraining loss, they may have extreme variation in other qualities, including specific benchmarks. This paper does not engage substantially with the existing literature on the connection or lack thereof between loss and specialized metrics; it nods to them in the literature review on "Relationship of Pre-training Loss and Task Performance" but does not clarify why their findings are so different from existing results.

>We argue that the perplexity of correct options is not the correct metric to evaluate the performance of multi-choice questions. The correct metric of multi-choice questions should reflect the ability of distinguishing correct options from incorrect options.

I looked into this appendix because I was intrigued as to why their results are so strongly at odds with Schaeffer et al. I completely disagree with this claim (which the paper provides no support for); if perplexity is the wrong metric, why measure loss at all? To the contrary, recent work including https://arxiv.org/pdf/2404.02418 and https://arxiv.org/pdf/2305.13264 continue to support the notion that claims of emergence should consider probabilities rather than exact matching, which obscures auxiliary task dependencies such as the capability of answering in multiple choice format. In general, our community now understands how much emergence can be attributed to metric thresholding effects, so you need much stronger evidence to argue against Schaeffer et al.

The core claim that one can predict performance on any given task just from the loss of a model would be a more useful finding if it applied across different architectures, not just different data and model scales. Although the paper considers both Llama and Pythia, they do not plot these different architectures on the same figures and it is therefore not clear whether loss is similarly predictive across architectures or whether it is only within an architecture, in which case there is no real argument for preferring it over scale as a basis of measurement. There is no excuse for continuing to present metric thresholding effects as evidence of meaningful emergence.

> That is, by ignoring the color differences (model sizes), the data points of different models are indistinguishable.

This simply doesn’t seem to be true. RACE in particular does seem to have a different slope for green compared to the other colors. For CLUESC orange and blue seem to be nearly separable, although that could be random. Still seems to be true for a bunch of these and it’s an interesting point to make, but I think you need to qualify it by actually comparing the slopes from a linear regression for each color.

> an ability is emergent if it is not present in language models with higher pre-training loss, but is present in language models with lower pre-training loss.

Not clear what it means to claim any ability is not present and then is present. It seems that they mean that below a certain point you have random chance answers. So by definition, if you perform at random chance at any point, then they would say the ability is emergent regardless of whether there is a clear breakthrough. This is the definition of a thresholding effect.

Minor:
- Please use natbib with \citet when you have an inline citation.
- "emergency trends" should be emergent
- > until the pre-training loss decreases to about 2.2, after which the performance gradually climbs as the loss increases.
    - Do you mean the loss decreases? Or am I missing something? Why would the loss increase after this point?

**Questions:**

Could you discuss a little further why your findings are so at odds with existing work like Schaeffer et al.? As far are as I can tell, it is because you've decided to define emergence without accounting for straightforward metric thresholding effects.

**Limitations:**

No obvious unlisted limitations.

---

> ### Author Rebuttal · Authors · 2024-08-04
>
> We want to argue that the reviewer misunderstands the difference between our work and Schaeffer et al. In fact, Schaeffer et al do not use perplexity as the evaluation metric for multi-choice tasks. Instead, they propose to use Brier Score, which we also evaluate in Figure 4. The main difference is that Schaeffer et al only evaluate the final checkpoints of different models while we also evaluate the intermediate checkpoints. With more data points available, we show that emergency on some tasks exists even with continuous metrics like Brier Score.
>
> We are against evaluating multi-choice tasks with the perplexity of correct options because we think a good metric should reflect the ability of distinguishing correct options from incorrect options. However, it is not the main argument of the paper and also not the reason why our findings are different from Schaeffer et al.
>
> We prefer loss over compute/scale since it can better track the learning dynamics of intermediate checkpoints. Loss of intermediate checkpoints is influenced by many factors beyond compute/model scale, such as learning rate scheduling. We show the performance-vs-compute curves in Figure 6 and find that points from different models do not fall on the same curves on most tasks.

---

> > ### Comment · Reviewer_BK9j · 2024-08-07
> > **Clarification**
> >
> > Thank you for clarifying, I confused your arguments against Xia et al. (that perplexity is the wrong metric) and against Schaeffer et al. (that Brier score is the wrong metric). I think this would have been easier to keep straight if you used the natbib \citet in your discussion, but I acknowledge that I confused those two different arguments, although you did make both of them.
> >
> > To be clear then, you claim that the same emergence point that Schaeffer et al. dismiss is actually a real point of emergence because the blue dots (1.5B) are flat whereas the orange dots (6B) have a positive slope. However, looking at each individual color makes it clear in the Brier score plot: for a 6B model, before the supposed phase transition, there is a lot more noise in the distribution of scores but they do not have a clearly insignificantly flatter trendline. In particular, it is entirely obvious that the 1.5B models over perform on the task relative to their loss when we compare them to 6B models with similar loss.
> >
> > The reason for the emergence artifact in the thresholding functions is clearly a thresholding effect. This fact persists even if you look at loss on the X axis. I appreciate the creativity of plotting different models together with all their checkpoints, but the defense of emergence in this paper does not seem to hold, and that appears to be the framing given. The argument that we can use earlier checkpoints to plot scaling laws for many models simultaneously does not hold unless those laws all show similar slopes for different models, and they obviously don't or this paper would have shown different architectures together on the same plot instead of just different scales of the same architecture, which would make the different slopes more obvious.
> >
> > I would raise my score if you could show that Llama and Pythia have similar slopes when plotted together and that they therefore support the idea that the 6B and 1.5B actually have the same slope, which they don't seem to as the 6B simply increases in variance of performance below the threshold.

---

> > > ### Author Response · Authors · 2024-08-08
> > >
> > > For the improvement on Brier Score before the emergency threshold, we already analyze this in Line 223 on Page 8. We find that the distribution of the correct answers is uniform in the four options in MMLU and C-Eval. A naive predictor that always gives uniform probability to all the options can achieve a Brier Score of 0.75, while another naive predictor that always gives all the probability to the first option has a Brier Score of 1.5. However, both models do not learn the relationship between questions and correct options at all. Therefore, decrease of Brier Score above 0.75 does not represent improvement on performance. In the plot we mark the Brier Score of random guess (0.75) in black dashed lines and obviously above the threshold the model's Brier Score is worse than random guess. We can't say a model's ability on a task is improving while its performance is worse than random guess.
> > >
> > > We can't plot the curves of Llama and Pythia on the same plot because they are trained on different corpora. Pythia is trained on Pile, of which webpages constitutes 18%. Llama is trained on their own corpora, of which webpages 67%. In other words, they do not have a common x-axis. Also, the intermediate checkpoints of Llama are not publicly accessible. Training models with different architectures is beyond available computational resources. However, we already show that models with the same architecture and different sizes show similar trends. We believe it will be helpful for the community since most models adopt Llama architecture with moderate modifications.

---

> > > > ### Comment · Reviewer_BK9j · 2024-08-08
> > > > **Differences from Schaeffer**
> > > >
> > > > My understanding reading the paper was that you find different results from Schaeffer et al. by rejecting the argument for continuous metrics. You stated
> > > >
> > > > >  The main difference is that Schaeffer et al only evaluate the final checkpoints of different models while we also evaluate the intermediate checkpoints. With more data points available, we show that emergency on some tasks exists even with continuous metrics like Brier Score.
> > > >
> > > > I don't believe that this is true if you look at the Brier score plots provided and note that the slopes are substantially different depending on which model you are considering. Therefore, you haven't provided evidence that intermediate training time can be treated similarly to other scale effects in terms of the relationship between aggregate loss and specific task capabilities (which makes sense, given that early stage training dynamics are different from later stage).
> > > >
> > > > The remaining difference from recent findings against emergence is therefore the rejection of continuous metrics, as I stated:
> > > >
> > > > > In general, our community now understands how much emergence can be attributed to metric thresholding effects, so you need much stronger evidence to argue against Schaeffer et al.
> > > >
> > > > While you make a fair point that you don't have access to each training corpus to allow evaluation of train loss and the models are trained on substantially different data, you then do not have the evidence to support an argument that aggregate (train) loss is directly predictive of specific task capabilities.

---

> > > > > ### Author Response · Authors · 2024-08-09
> > > > >
> > > > > > I don't believe that this is true if you look at the Brier score plots provided and note that the slopes are substantially different depending on which model you are considering.
> > > > >
> > > > > The slope of the 1.5B model on Brier Score is different from those of 6B and 32B models is because all the checkpoints of the 1.5B model performs no better than the random chance on MMLU and C-Eval, which means these points are below the tipping point of our proposed emergent abilities. For points at the intersection of 6B and 32B models, which are above the tipping point, the slopes are very close. Our point is that on some tasks with emergent abilities (like MMLU and C-Eval), the slopes of loss-performance curves at high-loss regions are different from those at low-loss regions. Therefore it is very difficult to predict the performance of larger models on these tasks with the performance of smaller models.

---

### Official Review · Reviewer_ZLu4 · 2024-07-09

**Soundness:** 4
**Presentation:** 3
**Contribution:** 4
**Rating:** 7
**Confidence:** 5

**Summary:**

They demonstrate that 1) pre-training loss is generally predictive of downstream capabilities in language models, rather than models size or number of tokens used during pre-training; and 2) emergent capabilities can also be clearly described in terms of pre-training loss. They also demonstrate that using continuous metrics, they still observe emergence, countering findings from prior work. They conduct their analysis on a suite of standard English and Chinese evals, using a range of models that they pre-trained themselves. They also further validate their findings using the llama and Pythia series of models.

**Strengths:**

* The paper does a good job of correcting an extremely prominent (but largely incorrect) narrative/claim that the phenomenon of emergence in LLMs always completely disappears when a continuous metric is used in place of a discontinuous one. This paper demonstrates a few cases where even in the presence of a continuous metric, emergence is still observed. Correcting this misconception in the scientific discourse is important.
* The paper is overall well written and the experiments are very thorough, encompassing a range of models and tasks.
* They even went out of their way to ablate the effect of learning rate schedule on their findings in Section 2.4, this is extremely thorough work, and they should be commended for it.

I have a handful of concerns regarding the writing, as discussed in the weaknesses section, but on balance I think this paper should be accepted, on the condition that my concerns are fixed in the camera ready.

**Weaknesses:**

* From the intro: "For example, LLaMA-13B with less compute [53] can outperform GPT-3 (175B) on MMLU [21]": Training for longer could be one explanation for this, another one could be llama used a higher quality corpus than gpt-3 did. I think the data quality element is a little underrated. In general we should expect different pretraining datasets to potentially have different loss / different emergence points (even if the x-axis loss is on a held out validation set and models have the same vocabulary). There is some discussion of this in the limitations section, but in general I think this point should be highlighted more in the paper. Concretely, the pretraining loss -> emergence point phenomenon is only guarenteed to be consistent for a series of models pretrained on the same data.
* The discussion of exact match in Section 2 kind of comes out no nowhere, and I'm not sure why it is discussed where it is. It would be great if there were more motivation for this in the paper.
* I understand why you did the ablation that you did in Section 2.4 (ablate the effect of lr schedule, as in Chinchilla), but in the paper this is not very clearly motivated and readers with less background knowledge might not understand why this ablation is important.
* “Note that there is one only exception at the early stage of LLaMA-65B. We can see that when the training loss is higher than 1.8, LLaMA-65B performs worse than smaller models with the same training loss.”: This could be because they used some exponential smoothing on either their loss or downstream performance plots. Exponential smoothing would perturb the earlier points more than other points (especially for curves that are fast changing or have a rapid change at the beginning), potentially leading to this effect. Moreover, did you smooth out the loss in some way when plotting loss verses downstream in the previous sections? If so, this would at least be good to note in the paper.
* Section 3.1 could use clearer discussion of explanations for why emergence occurs on the different tasks. I think grokking is a possible explanation, but what exactly grokking means (outside of the context of the simple algorithmic tasks presented in the original grokking paper) is a little vague/imprecise, and I also don't think this is the only explanation. In the case of gsm8k, emergence is, in my opinion, more likely due to the fact that the model has to get a sequence of reasoning steps correct to answer the question, this leads to an exponential (e.g. p(step_correct)^n_steps) for getting the full answer correct (some analysis of this in the paper could be cool). In the case of MMLU and C-Eval there actually is an existing paper which discovers an explanation, using interpretability techniques, for emergence on multiple choice tasks (https://arxiv.org/pdf/2307.09458).
* There's a handful of related works that aren't discussed in the paper and whose findings contradict those of this paper (I'll give you my reasoning for why your work might not contradict theirs, as a suggestion in parentheses, but you should include some discussion of this in the paper somewhere, either in the related work or elsewhere):
1) https://proceedings.mlr.press/v202/liu23ao.html (my understanding is that their theory only applies near the global optim, which is generally not the case with real-world language models)
2) https://arxiv.org/pdf/2109.10686 (see the paragraph "zooming in verses zooming out", your paper is more about the zoomed out setting)
3) Not strictly language modeling, but see Figure 3 of https://cdn.openai.com/papers/Generative_Pretraining_from_Pixels_V2.pdf (could be because they are doing probing so hidden-state size increases probe capacity causing differences with model size, but it's unclear if this is actually the case)

**Questions:**

See the weaknesses for most of my questions/concerns.

**Limitations:**

No major limitations. See the weaknesses section.

---

> ### Author Rebuttal · Authors · 2024-08-06
>
> We agree with the influence of data quality on performance besides compute. We will add the point in the introduction when making the comparison.
>
> Thank the reviewer for appreciating the ablation of learning rate schedule. We will make it clearer in the final version.
>
> The exponential smoothing is one possible explanation for the outliers on LLaMA-65B. We will add the explanation to the paper. Also, we didn't use any smoothing in all the plots in the paper.
>
> Thank the reviewer for providing related works on explanations of emergent abilities. We will add these explanations in the paper. For contradictory conclusions of previous works [1][2], we think they mainly study the pretraining-finetuning paradigm, in which inductive bias helps improve transferability. Instead, we study the pretraining-prompting paradigm without finetuning on specific tasks, which is more common in language model pretraining. We will add more discussion about this in the paper.

---

> > ### Comment · Reviewer_ZLu4 · 2024-08-13
> >
> > Thanks, I appreciate the response. I think this is a good work and will keep my score.

---

### Official Review · Reviewer_NWes · 2024-07-12

**Soundness:** 3
**Presentation:** 3
**Contribution:** 3
**Rating:** 6
**Confidence:** 4

**Summary:**

The paper investigates the link between the pre-training loss of LLMs and their downstream performance on popular benchmarks. The authors train a series of models ranging from 300M to 32B parameters on English-Chinese datasets of different sizes and study how these models perform on TriviaQA, HellaSwag, RACE, WinoGrande, MMLU and GSM8K benchmarks and their Chinese counterparts, as a function of their pre-training loss. They observe a strong link between pre-training loss and benchmark performance that is not affected by model size and dataset size, from which they conclude that pre-training loss is the main indicator of downstream performance. On MMLU and GSM8K (and their Chinese counterparts), models exhibit an emergent behavior wrt. loss, i.e. their performance is close to random before a loss value of about 2.2 and steadily increases for lower loss values. This relationship seems to hold, even under continuous evaluation metrics.

**Strengths:**

1. Formulating downstream (emergent) abilities in LLMs in terms of pre-training loss is a valuable contribution, that can unify several different factors that are so far thought to contribute to emergence, such as pre-training compute, model size, and (test-)perplexity. This perspective also provides a better connection between scaling laws -- which typically describe the relationship between parameter count, dataset size, compute, and pre-training loss -- and emergent abilities, which mostly have been studied through the lens of parameter count and compute so far.
2. The paper presents an extensive evaluation over a large range of model sizes, dataset sizes, compute budgets, that show a strong connection between pre-training loss and downstream performance. The results are additionally validated with results and models proposed by prior work (Llama-1 and Pythia).
2. The paper is easy to understand and follow.

**Weaknesses:**

1. The paper seems to make two main claims: 1) that pre-training loss is the main predictor of downstream performance, and 2) that emergent abilities appear suddenly after models cross a certain loss-threshold. I find the evidence for 1) to be convincing, but I am somewhat less confident about 2). This is mostly because the paper only shows emergent abilities on a subset of the benchmarks they were originally proposed on (MMLU and GSM8K) in [1]. Notably, BigBench is absent. Including results on the tasks in BigBench that were shown to exhibit emergent behavior would be helpful to judge whether pre-training loss predicts the emergence threshold as well as pre-training compute or model size. Including these benchmarks would also help to put the paper into a better perspective wrt. subsequent work questioning the existence of emergent abilities [2], which also studies these benchmarks.
2. There are some discrepancies and unexplored links between the results reported in the paper and findings in prior work. To develop a better understanding of how the findings here relate to prior work, it would be helpful to include a discussion in the paper. See Questions 2. and 3. below for more details.
3. I find the paragraph in 3.1, line 197 rather speculative. Grokking refers to an improvement on the test-set, despite stagnation on the training set, whereas the models studied here seem to still be improving on the training set. I am not sure whether there is a link between grokking and the emergent behavior on the specific tasks here.
4. Minor: There are a number of typos and small writing issues, e.g.
    - "has thus far less studied" (line 23), ", a highly over-parameterized models" (line 79), "coefficients how that" (line 126),...
    - line 120: "climbs as the loss increases" should probably be "decreases".
    - line 141: "On each line" seems to be missing a reference to Figure 2.

References
- [ 1] Emergent Abilities of Large Language Models, Wei et al., https://arxiv.org/abs/2206.07682
- [ 2] Are Emergent Abilities of Large Language Models a Mirage?, Schaeffer et al., https://arxiv.org/abs/2304.15004

**Questions:**

1. The paper claims that pre-training (test) loss is a good indicator of downstream performance. What is the composition and size of the test set that the loss is evaluated on? What properties (size, diversity, etc.) should such a test-set have in order to be predictive of downstream performance?
2. Prior work [1, 2] has found that emergent behaviors can become predictable with "higher resolution" benchmarks, i.e. with larger test sets or repeated sampling from the model. This is a dimension that the paper does not touch upon. Do the authors believe, that the studied emergent abilities would still appear only after models cross the particular loss threshold, i.e. be essentially 0 before, even with those higher resolution benchmarks?
3. In related work (line 254), the authors mention that some prior work [3] has observed a disconnect between pre-training loss and downstream performance, which stands in contrast to the claims made in the paper. It would be great if the authors could comment on more on the reasons for these discrepancies.

References
- [ 1] Are Emergent Abilities of Large Language Models a Mirage?, Schaeffer et al., https://arxiv.org/abs/2304.15004
- [ 2] Predicting Emergent Abilities with Infinite Resolution Evaluation, Hu et al., https://arxiv.org/abs/2310.03262
- [ 3] Same Pre-training Loss, Better Downstream: Implicit Bias Matters for Language Models, Liu et al., https://proceedings.mlr.press/v202/liu23ao.html

**Limitations:**

The paper sufficiently discusses limitations, in my opinion.

---

> ### Author Rebuttal · Authors · 2024-08-06
>
> For weakness 1, the results on BIG-Bench are presented in the PDF of global rebuttal. The original emergent ability paper evaluates 4 tasks on BIG-Bench. The test set size of the figure-of-speech detection task is too small and the variance is too high. Therefore we evaluate the other three tasks.
>
> For question 2: We already study the effect of continuous metrics in section 3.2, in which we evaluate MMLU and C-Eval with two continuous metrics, one of which is the Brier Score used in Schaeffer et al. The results show that continuous metrics cannot eliminate the observed tipping point in the performance curves. Increasing the size of the test set of GSM8K is not straight-forward. We will try repeated sampling from the model, but it takes some time. We think repeated sampling does not change the results since it only decrease the variance of the performance evaluation.
>
> For question 3: Liu et al. mainly analyze BERT-like models, which are pretrained on masked language modeling and finetuned and evaluated on supervised classification tasks, which we denote as "transfer learning setting" in the paper. Implicit bias in model sizes and training algorithms can change the transferability of pretrained knowledge. Our work focuses on GPT-like models, which are pretrained on autoregressive language modeling and evaluated on prompted tasks without finetuning. Since the pretraining and evaluation settings are more consistent, the transferability is less important.
>
> For weakness 3, pretraining is generally considered as multi-task learning. It is possible that the model already stagnates on some tokens in the pretraining corpus, such as digit calculation, while the overall pre-training loss is still decreasing.

---

> ### Comment · Reviewer_NWes · 2024-08-13
>
> I thank the authors for answering my questions in the rebuttal, and for providing additional results on BIG-Bench. They help in assessing how the findings of this work relate to observations made in prior work and make me more confident about the results.
>
> I continue to believe that the pre-training loss perspective is a valuable contribution for predicting emergent downstream abilities. However, I am still not quite sure that the downstream abilities emerge "suddenly". The results on BIG-Bench, particularly word unscramble and IPA transliterate, might actually be following a continuous trend, e.g. a sigmoid, and for modular arithmetic there seem to be too few data points below the "emergence threshold" to accurately judge whether there is a sudden transition. Some sort of higher-resolution sampling might also show more continuous trends here.
> Therefore, I maintain my score.
>
> An interesting suggestion for future work would be to compare checkpoints from different model families on the same test-set (potentially with tokenizer-based normalization) to see whether the same loss values result in the same downstream performance for across families. To do this it would be possible to use open-checkpoint models such as Pythia, LLM360 [1] and OLMo [2].
>
> - [1] LLM360: Towards Fully Transparent Open-Source LLMs, Liu et al., https://arxiv.org/abs/2312.06550
> - [2] OLMo: Accelerating the Science of Language Models, Groeneveld et al., https://arxiv.org/abs/2402.00838

---

> > ### Author Response · Authors · 2024-08-13
> >
> > Thank you for the valuable feedback. We understand that on some tasks it is difficult to differentiate emergent abilities from exponential performance curves. We want to emphasize that the point of emergent abilities is to identify some tasks on which it is difficult to predict the performance of larger models with the performance of smaller models. On word unscramble and IPA transliterate, before the tipping point, the curve is speciously linear with a very small slope, rather than exponential. Therefore it is very difficult to predict the sudden increase in the performance as the pretraining loss further decreases. This also distinguishes these tasks from those whose performance increases smoothly with an almost constant slope.

---

> ### Comment · Reviewer_NWes · 2024-08-14
>
> I thank the authors for their clarification. I still believe that some of the transitions may potentially be predictable from higher loss models.
>
> I will maintain my score, but increase my confidence level, and would support acceptance of the paper.

---

### Official Review · Reviewer_RA4k · 2024-07-13

**Soundness:** 2
**Presentation:** 2
**Contribution:** 2
**Rating:** 4
**Confidence:** 3

**Summary:**

This paper investigates emergent abilities in language models from the perspective of pre-training loss, rather than model size or training compute. The authors challenge recent skepticism about emergent abilities by demonstrating that: (1) Models with the same pre-training loss, regardless of model and data sizes, exhibit similar performance on various downstream tasks. (2) Emergent abilities manifest when a model's pre-training loss falls below a specific threshold, before which performance remains at random guessing levels.
The study examines different metrics (including continuous ones) and proposes a new definition of emergent abilities based on pre-training loss. The authors argue that this perspective better represents the learning status of language models and provides a more precise characterization of when emergent abilities appear during training.

**Strengths:**

- The paper offers a fresh approach to understanding emergent abilities by focusing on pre-training loss rather than model size or compute, providing valuable insights into the scaling behavior of language models.
- The paper builds upon existing scaling laws and provides a mathematical formulation (Equation 5) that explains the relationship between model size, pre-training loss, and emergent abilities.
- The study directly addresses recent challenges to the concept of emergent abilities, providing a nuanced perspective that reconciles conflicting observations in the field.

**Weaknesses:**

- The authors acknowledge that they have not considered fundamentally different model architectures (e.g., routed Transformers) or non-Transformer architectures. This limitation may affect the generalizability of their findings.
- As noted in the limitations section, pre-training loss is affected by tokenizers and pre-training corpus distribution, making direct comparisons between models trained on different corpora challenging. While the authors suggest using normalized perplexity on a public validation set, this solution is not implemented in the current study.
-  While the paper establishes a correlation between pre-training loss and emergent abilities, it does not provide a causal explanation for why certain abilities emerge at specific loss thresholds. A deeper investigation into the underlying mechanisms could strengthen the paper's contributions.
- While the authors discuss some contrary observations from previous studies, a more comprehensive comparison with existing literature on emergent abilities and their proposed explanations would enhance the paper's positioning within the field.

**Questions:**

None.

**Limitations:**

Yes.

---

> ### Author Rebuttal · Authors · 2024-08-04
>
> We admit the limitations of the work in different model architectures, tokenizers, and pre-training corpus distribution. The main reason is that the work analyzes not only the final checkpoints of different models, but also the intermediate checkpoints, which are not publicly available for many open-source models. Training multiple models with different architectures, tokenizers, and pre-training corpus is beyond our available computational resources. We also want to argue that previous works on scaling laws often study a fixed choice of architectures, tokenizers, and pre-training corpus [1][2].
>
> We also admit the lack of theoretical analysis in the paper. The main question the paper answers is whether the emergent abilities exist and how to track them, which is necessary for further study on emergent abilities. We believe the paper can inspire more theoretical work on the topic.
>
> 1. Kaplan et al. Scaling Laws for Neural Language Models.
> 2. Hoﬀmann et al. Training Compute-Optimal Large Language Models.

---

### Author Rebuttal · Authors · 2024-08-06

The results on BIG-Bench are presented in the PDF file. The original emergent ability paper evaluates 4 tasks on BIG-Bench. The test set size of the figure-of-speech detection task is too small and the variance is too high. Therefore we evaluate the other three tasks. We can observe with pretraining loss as the x-axis, we can clearly observe the tipping point in the performance (compared with Figure 2 in Wei et al.)

---

### Decision · Program_Chairs · 2024-09-25

**Decision:**

Accept (poster)

**Comment:**

This work demonstrates that 1) pre-training loss is generally predictive of downstream capabilities in language models, rather than model size or number of tokens used during pre-training; and 2) emergent capabilities can also be clearly described in terms of pre-training loss. They also demonstrate that using continuous metrics, they still observe emergence, countering findings from prior work. They conduct their analysis on a suite of standard English and Chinese evals, using a range of models that they pre-trained themselves. They also further validate their findings using the Llama and Pythia model families.

I am not going to factor in the review of Reviewer RA4k, as it is too short and the reviewer did not engage further either in discussions with the authors or discussions with the other reviewers.

This work has the potential to be a timely contribution on understanding emergent properties in LLMs, if the results support the conclusions drawn by the authors. All reviewers other than ZLu4 were initially skeptical at least to some degree about the evidence that the authors provide for one of the two major claims in this work -- that a specific pre-training loss threshold is needed to lead to emergent abilities in LMs. The authors had lengthy exchanges with the reviewers and most reviewers were satisfied by the author response. There are still some outstanding concerns from Reviewer BK9j, which I urge the authors to take seriously, specifically regarding whether all modes indeed lie on the same general curve, especially because there is little overlap between the loss ranges for different model size. Despite this remaining concern, there is strong support for this work from the remaining reviewers, and I believe this work will at least serve to deepen the discussion on the emergent properties of LLMs. I suggest that the authors incorporate the reviewers' suggestions in a revised manuscript, particularly emphasizing the role of pretraining data and clarifying the comparison with and discussion of previous work.